# Empowering Federated Graph Rationale Learning with Latent Environments

## ABSTRACT

The success of Graph Neural Networks (GNNs) in graph classification has heightened interest in explainable GNNs, particularly through graph rationalization. This method aims to enhance GNNs explainability by identifying subgraph structures (i.e., rationales) that support model predictions. However, existing methods often rely on centralized datasets, posing challenges in scenarios where data privacy is crucial, such as in molecular property prediction. Federated Learning (FL) offers a solution by enabling collaborative model training without sharing raw data. In this context, Federated Graph Rationalization emerges as a promising research direction. However, in each client, the rationalization methods often rely on client-specific shortcuts to compose rationales and make task predictions. Data heterogeneity, characterized by non-IID data across clients, exacerbates this problem, leading to poor prediction performance. To address these challenges, we propose the Environment-aware Data Augmentation (EaDA) method for Federated Graph Rationalization. EaDA comprises two main components: the Environment-aware Rationale Extraction (ERE) module and the Local-Global Alignment (LGA) module. The ERE module employs prototype learning to infer and share abstract environment information across clients, which are then aggregated to form a global environment. This information is used to generate counterfactual samples for local clients, enhancing the robustness of task predictions. The LGA module uses contrastive learning methods to align local and global rationale representations, mitigating performance degradation due to data heterogeneity. Comprehensive experiments on benchmark datasets demonstrate the effectiveness of our approaches. Code is available at https://anonymous.4open.science/r/Codes-of-EaDA-48DB/.

**ACM Reference Format:**
Anonymous Author(s). 2025. Empowering Federated Graph Rationale Learning with Latent Environments. In *Proceedings of the ACM Web Conference 2025 (WWW '25), April 28 - May 2, 2025, Sydney, Australia.* ACM, New York, NY, USA, 11 pages. https://doi.org/XXXXXXX.XXXXXXX

## 1 INTRODUCTION

The recent success in Graph Neural Networks (GNNs) for graph classification tasks have catalyzed significant interest in explainable GNNs [11, 14, 33, 39, 44]. Among them, graph rationalization [34, 47] has garnered considerable attention. The objective of graph

rationalization is to improve the explainability of GNNs by identifying a subgraph structure, referred to as the rationale, which supports the model's prediction results. For example, in molecular property prediction, in judging whether *Glutamic Acid* $(C_5H_9NO_4)$[1] is slightly soluble in water, we extract the carboxy substructure $(-COOH)$ as the rationale to support this prediction.

The reliance of existing graph rationalization methods on centralized datasets presents a misalignment with many critical graph classification scenarios, particularly those where data privacy is important. For instance, in molecular property prediction tasks, academic institutions and pharmaceutical companies are often reluctant to share proprietary molecular datasets due to the intrinsic value of intellectual property and chemical data. Federated Learning (FL) [27, 42, 48] offers a promising avenue to address this challenge. FL is a decentralized machine learning framework that enables multiple clients to collaboratively train local models, with only model parameters being aggregated via a central server to form a global model, thereby eliminating the need to exchange raw data.

Therefore, Federated Graph Rationalization emerges as a valuable research direction. Commonly, for a vanilla federated graph rationalization method (Fed-vanillaGR), in each client, it first employs a *rationale extractor* to identify sufficient nodes (i.e., the subgraph) within the graph and then generate the corresponding node representations. A *predictor* then produces the task results based exclusively on the representations of these identified nodes. Finally, the recognized subgraphs serve as the rationale supporting the prediction results. On the server side, it aggregates the parameters of the *rationale extractor* and *predictor* from each client and distributes the aggregated parameters back to each client, thereby completing the training of Fed-vanillaGR.

However, one of the primary obstacles in this direction is data heterogeneity [41, 46]. Specifically, data heterogeneity refers to the non-independent and identically distributed (non-IID) nature of cross-client data in FL settings. This variation arises from factors such as differences in data collection methodologies across clients, leading to distinct environmental contexts (i.e., differing data distributions) at each client. In FL scenarios, data heterogeneity may exacerbate the problem that current graph rationalization methods are prone to exploit shortcuts for task prediction [2, 39]. Specifically, for each client, the rationalization methods may leverage client-specific shortcuts to make predictions. Among them, shortcuts exhibit correlations with the task results but lack a causal relationship, commonly referred to as spurious correlations. Spurious correlations (e.g., statistical correlation) are influenced by the environment where the data resides, and alterations in this environment can lead to the changes of spurious correlations, further result in the prediction errors. Due to the distinct environments

---

[1] $C_5H_9NO_4$:

of each client, the client-specific shortcuts learned are also different. When local rationalization models are aggregated into a global model, such inconsistencies may lead to significant performance differences compared to models trained on centralized datasets.

To address this problem, Yue et al. [46] propose the FedGR method, which leverages data augmentation methods to mitigate the effects of data heterogeneity. Their approach assumes that the environments across clients are unavailable. Then, by exploiting the differences between the global and local models, they generate shortcut conflicted data samples that do not conform to the current client environment. Despite showing promising results, practical applications of FedGR have revealed significant efficiency bottlenecks. For example, training time is approximately 5 times longer compared to the vanilla federated graph rationalization. A potential reason for this inefficiency is that FedGR's reliance on unavailable environment assumptions necessitates the training of an additional model to capture the differences between global and local models, which then supports data augmentation. Then additional model reducing the training efficiency ultimately. Considering that computational resources in several clients may be limited in real-world FL scenarios, the practical adoption of FedGR may be unavailable. Therefore, we argue that this "*data augmentation pattern*" can be further explored to improve the federated graph rationalization.

Along this research line, in this paper, different from previous methods that assume the latent environment is unavailable, we assume that the latent environment can be inferred and propose an *E*nvironment-*a*ware *D*ata *A*ugmentation (EaDA) method for Federated Graph Rationalization. This method comprises two key components: the Environment-aware Rationale Extraction (ERE) module and the Local-Global Alignment (LGA) module. Specifically, in the ERE module, we recognize that the environments of different client data vary. We initially employ a prototype learning approach to infer the potential environment of each client, which is then uploaded to the server. The uploaded environment information, being abstracted prototype data, preserves the privacy of the dataset. Upon collecting this information from each client, the server merges the environment data to construct the global environment information (assuming $N$ clients with $T$ environments each, the final number of merged global environment is $N \times T$), which is subsequently distributed to all clients. Once clients receive the global environment information, we utilize an environment-aware generator to map samples from the current environment to other environments, thus creating new counterfactual samples. Importantly, as the environment does not affect task predictions, the labels of the generated samples remain unchanged. By incorporating both original and generated samples during model training, we can derive more faithful task results. Additionally, to further mitigate the data heterogeneity problem, the LGA module employs a contrastive learning approach to align the global rationale with the local rationale subgraph representations. This collaborative learning strategy allows local models to access global information, thereby alleviating the performance degradation caused by data heterogeneity. Consequently, more robust model parameters are provided during model aggregation, resulting in a global model with enhanced performance. Experiments over real-world benchmarks [18, 21] and various synthetic datasets [39] validate the effectiveness of EaDA.

## 2 RELATED WORK

### 2.1 Graph Rationalization.

The remarkable advancements of Graph Neural Networks (GNNs) [10, 12, 13, 25, 36] have catalyzed significant interest in the explainability of graph classification tasks [3–6, 22]. Within this domain, graph rationalization methods have emerged as a focal point. However, as demonstrated by [2], graph rationalizations are prone to exploiting shortcuts in data for prediction. Therefore, current approaches primarily focused on how to compose faithful rationales and further address the shortcut problems. For instance, DIR [39] introduced a methodology for identifying invariant rationales by disentangling input graphs into rationale and non-rationale subgraphs. They treated non-rationale subgraphs as distinct environments, combining them with rationales to generate counterfactual samples for prediction. Building on this, subsequent methodologies [9, 22, 24, 34] have been developed. Unlike DIR, which treated each non-rationale graph as an environment, GIL [22] and C2R [47] inferred local environments by clustering non-rationale representations within a mini-batch. Similarly, HSE [29] and EQuAD [43] employed environment inference techniques to determine environment labels for each sample. Additionally, some studies focused on restructuring rationalization methods to mitigate shortcut issues. For example, DARE [45] introduced a self-guided rationalization framework that captures comprehensive input information through a disentanglement approach. GSAT [28] integrated information bottleneck theory into the rationalization framework using a learned stochasticity-reduced attention mechanism. DIVE [35] employed subgraph diversity regularization to enhance variation in the rationale patterns identified by models.

While rationalization methods have been extensively studied in centralized datasets, their application in FL scenarios remains underexplored. FedGR [46] present the first federated graph rationalization method, leveraging the difference between global and local models in FL to design difference-aware data augmentation techniques. This approach can generate anti-shortcut data samples for each client, thereby enhancing the effectiveness of rationalization methods in FL scenarios.

### 2.2 Federated Learning.

Federated Learning (FL) algorithms have garnered significant attention due to their capacity to address data security and privacy concerns [26, 27, 37, 40, 42]. Among these algorithms, [30] introduced a knowledge transfer method that leveraged actively selected small public data to transfer high-quality knowledge within FL frameworks while ensuring privacy guarantees. This approach represented a significant advancement in maintaining data privacy without compromising on the quality of the learned models. Additionally, [31] proposed a selective knowledge sharing mechanism for federated distillation, designed to identify and share accurate and precise knowledge derived from local and ensemble predictions. Recent research has also focused on eliminating spurious correlations in training data, which can lead to biased models. For instance, [8] proposed a FL framework aimed at mitigating spurious correlations and preventing models from becoming biased towards specific demographic groups. This framework addressed fairness in model training, a critical aspect in deploying machine

learning systems in diverse and sensitive applications. Similarly, [41] introduced a bias-eliminating augmentation method within the FL setting. By identifying and incorporating desirable causal and shortcut attributes into augmented samples, this method aimed to reduce spurious correlations and enhance the reliability of the trained models.

## 3 PRELIMINARIES

### 3.1 Problem Formulation

In this subsection, we delineate a rigorous formalization of the graph rationalization within FL scenarios. We consider a federated setting consisting of $N$ clients, denoted as $\{C_1, C_2, \ldots, C_N\}$. Each client has its own local graph datasets $\{\mathcal{D}_1, \mathcal{D}_2, \ldots, \mathcal{D}_N\}$. It is imperative to acknowledge the inherent diversity in data distributions across these clients, underscoring the variability among them.

For each client $C_k$, every graph-label pair is encapsulated as $(G_k, Y_k) \in \mathcal{D}_k$, where $G_k = (\mathcal{V}, \mathcal{T})$. Here, $\mathcal{V}$ denotes the set of nodes while $\mathcal{T}$ signifies the set of edges. The local task of graph rationalization involves a two-fold objective. Primarily, it entails the acquisition of a mask variable $\mathbf{M}_k \in \mathbb{R}^{|\mathcal{V}|}$ through a rationale extractor function $f_{s_k}(G_k)$, alongside the derivation of node representations $\mathbf{H}_{G_k} \in \mathbb{R}^{|\mathcal{V}| \times d}$. Subsequently, the rationale subgraph representation is computed via element-wise multiplication of the mask variable and the node representations, denoted as $\mathbf{M}_k \odot \mathbf{H}_{G_k}$. Finally, a predictor $f_{p_k}(\mathbf{M}_k \odot \mathbf{H}_{G_k})$ is trained to furnish accurate predictions.

The learning process revolves around the optimization of the extractor function $f_{s_k}^*(\cdot)$ and the predictor function $f_{p_k}^*(\cdot)$, minimizing the cross-entropy loss $\ell(\cdot)$ over the graph-label pairs in the client's dataset $\mathcal{D}_k$:

$$f_{s_k}^*(\cdot), f_{p_k}^*(\cdot) = \arg\min_{f_{s_k}, f_{p_k}} \mathbb{E}_{(G_k, Y_k) \sim \mathcal{D}_k} \left[ \ell \left( f_{p_k} \left( f_{s_k}(G_k) \right), Y_k \right) \right].$$

With a total of $T$ communication rounds, the overarching aim of rationalization at the global level is to derive the rationale extractor and predictor that fulfill the model aggregation process (i.e., **Model Aggregation** in Figure 1):

$$\hat{\Theta}^s = \sum_{k=1}^{N} \frac{|\mathcal{D}_k|}{\sum_{j=1}^{N} |\mathcal{D}_j|} \Theta_k^s, \quad \hat{\Theta}^p = \sum_{k=1}^{N} \frac{|\mathcal{D}_k|}{\sum_{j=1}^{N} |\mathcal{D}_j|} \Theta_k^p, \quad (1)$$

where $\hat{\Theta}^s$ represents the parameters of the global extractor $f_s(\cdot)$, $\hat{\Theta}^p$ represents the parameters of the global predictor $f_p(\cdot)$. Conversely, $\Theta_k^s$ denotes the parameters of the extractor $f_{s_k}(\cdot)$ in client $C_k$, and $\Theta_k^p$ denotes the parameters of the predictor $f_{p_k}(\cdot)$.

### 3.2 Vanilla Federated Graph Rationalization

In this subsection, we present the details of vanilla federated graph rationalization (Fed-vanillaGR), encompassing both the rationale extractor and the predictor components.

#### 3.2.1 Rationale Extractor in Fed-vanillaGR.
For each client $C_k$, given $(G_k, Y_k) \in \mathcal{D}_k$, the process of generating rationales within the rationale extractor $f_{s_k}(\cdot)$ entails a meticulous sequence of steps. Initially, an encoder $\text{GNN}_m(\cdot)$ orchestrates the transformation of each node in graph $G_k$ into a $d$-dimensional vector. Concurrently,

the extractor orchestrates the prediction of a probability distribution for the selection of each node as part of the rationale. This distribution is denoted as:

$$\tilde{\mathbf{M}}_k = \text{softmax}\left(W_m\left(\text{GNN}_m(G_k)\right)\right),$$

where $W_m \in \mathbb{R}^{2 \times d}$ denotes a weight matrix.

Subsequently, the extractor samples binary values (0 or 1) from the distribution $\tilde{\mathbf{M}}_k = \left\{\tilde{m}_k^i\right\}_{i=1}^{|\mathcal{V}|}$ to yield the mask variable $\mathbf{M} = \left\{m_k^i\right\}_{i=1}^{|\mathcal{V}|}$. To ensure differentiability during sampling, the Gumbel-softmax method [19] is employed:

$$m_k^i = \frac{\exp\left(\left(\log\left(\tilde{m}_k^i\right) + q_k^i\right)/\tau\right)}{\sum_t \exp\left(\left(\log\left(\tilde{m}_k^t\right) + q_k^t\right)/\tau\right)},$$

where $\tau$ denotes a temperature parameter, $q_k^i = -\log\left(-\log\left(u_k^i\right)\right)$, and $u_k^i$ is randomly sampled from a uniform distribution $U(0, 1)$.

Following this, an additional GNN encoder, $\text{GNN}_G$, is employed to extract the node representation $\mathbf{H}_{G_k}$ from graph $G_k$. The rationale node representation is formulated as the element-wise product of the binary rationale mask $\mathbf{M}_k$ and the node representation $\mathbf{H}_{G_k}$, articulated as $\mathbf{M}_k \odot \mathbf{H}_{G_k}$. Similarly, the complement node representation is computed as $(1 - \mathbf{M}_k) \odot \mathbf{H}_{G_k}$, signifying the nodes constituting the non-rationale.

#### 3.2.2 Predictor in Fed-vanillaGR.
The predictor $f_{p_k}(\cdot)$ encompasses a readout function and a classifier. Initially, the readout function is employed to derive the graph-level rationale $\mathbf{h}_{r_k}$ and complement $\mathbf{h}_{e_k}$ (i.e., the non-rationale) subgraph representations:

$$\mathbf{h}_{r_k} = \text{READOUT}(\mathbf{M}_k \odot \mathbf{H}_{G_k}),$$
$$\mathbf{h}_{e_k} = \text{READOUT}((1 - \mathbf{M}_k) \odot \mathbf{H}_{G_k}).$$

Finally, the classifier $\Phi(\cdot)$ yields the task results solely based on the rationale subgraphs:

$$\hat{Y}_{r_k} = \Phi\left(\mathbf{h}_{r_k}\right), \quad \mathcal{L}_r^k = \mathbb{E}_{(G_k, Y_k) \sim \mathcal{D}_k}\left[\ell(\hat{Y}_{r_k}, Y_k)\right]. \quad (2)$$

#### 3.2.3 Training and Inference.
During training, a sparsity constraint is imposed on the probability $\mathbf{M}_k$ of being selected as a rationale, as proposed in [24], to achieve a controlled level of sparsity in the generated rationale subgraphs:

$$\mathcal{L}_{sp}^k = \left| \frac{1}{|\mathbf{M}_k|} \sum_{i=1}^{|\mathbf{M}_k|} m_k^i - \alpha \right|, \quad (3)$$

where $\alpha \in [0, 1]$ is a predefined sparsity level. The overarching objective of Fed-vanillaGR in each client $C_k$ is expressed as:

$$\mathcal{L}_{rat}^k = \mathcal{L}_r^k + \lambda_{sp} \mathcal{L}_{sp}^k. \quad (4)$$

Upon completion of training by each client, the parameters of the extractor and predictor are transmitted to the server, which utilizes Eq(1) for parameter aggregation and distribution to finalize the training process.

During inference, $\mathbf{h}_r$ from the global server is employed to derive task results.

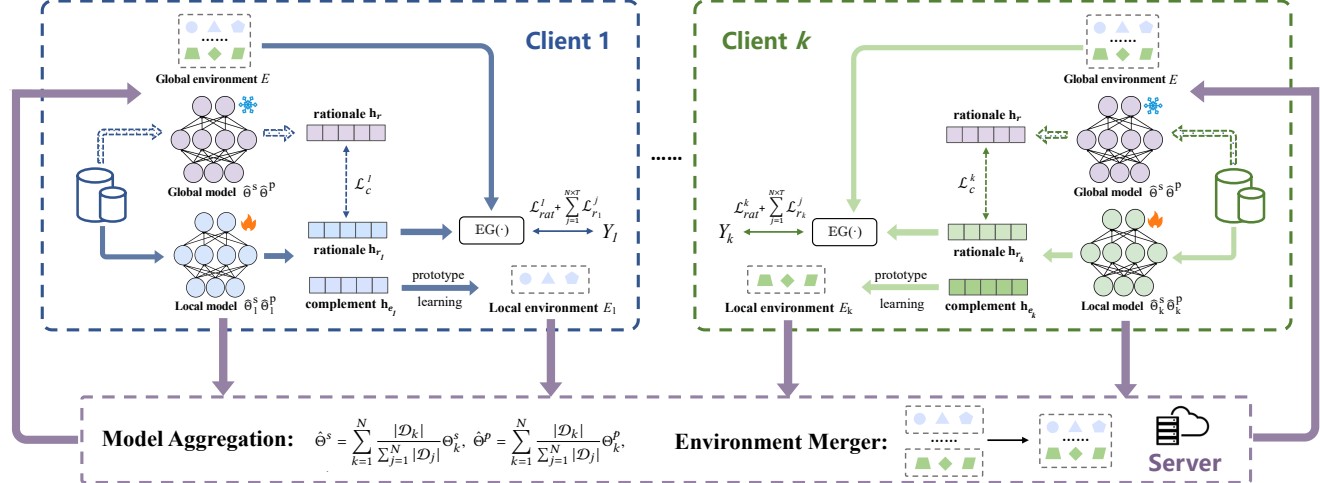

**Figure 1: The overall framework of EaDA. In each client, the solid lines indicate the process of the Environment-aware Rationale Extraction module, and the dashed represent the process of the Local-Global Alignment module.**

## 4 ENVIRONMENT-AWARE DATA AUGMENTATION FOR FEDERATED GRAPH RATIONALIZATION

Although Fed-vanillaGR provides a feasible solution for exploring the explainability of GNNs in FL scenarios, it still suffers from the data heterogeneity and local shortcut problem, degrading the effectiveness of Fed-vanillaGR. Therefore, in this section, as shown in Figure 1, based on the Fed-vanillaGR framework, we further propose an Environment-aware Data Augmentation (EaDA) for Federated Graph Rationalization method, consisting of an environment-aware rationale extraction module and a local-global alignment module.

### 4.1 Environment-aware Rationale Extraction

To mitigate the local shortcut problem, a logical approach is to introduce a global environment to each client, thereby generating more counterfactual samples and disrupting the spurious correlations inherent in the local dataset by altering the data environment of the training set. However, observing and obtaining the environment pose considerable challenges. Hence, we initially propose a prototype learning-based method for inferring the environment. Subsequently, the environment inferred from each client is transmitted to the server for aggregation, thereby obtaining global environment information, which is then disseminated. Finally, the data from the clients are mapped from their current environment to the global environment, facilitating the synthesis of counterfactual data through environment-aware generation to alleviate the local shortcut problem.

*4.1.1 Prototype learning-based Environment Inference.* After deriving the rationale subgraph and its complement, we proceed to infer the environment $E$. Specifically, as the complement subgraph encapsulates the correlation of variances across different distributions, which are indicative of environment-discriminative features, we leverage it to infer potential environments. Utilizing the concept of prototype learning [7, 32], we generate several prototype

embeddings for the complement subgraphs, defining these embeddings as the environment information. In practical implementation, within client $C_k$, given a batch $\left\{\left(G_k{}^i, Y_k{}^i\right)\right\}_{i=1}^B$ and the corresponding rationale and complement representations $\left\{\left(\mathbf{h}_{r_k}^i, \mathbf{h}_{e_k}^i\right)\right\}_{i=1}^B$, we compute the prototype embeddings (i.e., the environment information) as follows:

$$E_k = \text{Prototype}\left(\{\mathbf{h}_{e_k}^i\}_{i=1}^B\right), \tag{5}$$

where $E_k = \{\mathbf{e}_k^1, \mathbf{e}_k^2, \dots, \mathbf{e}_k^T\}$, and we utilize the k-means clustering algorithm [15, 22] as the Prototype($\cdot$) function in this study. Subsequently, we transmit the inferred environments from each client to the server and merge them to obtain the global environment $E = \{\mathbf{e}^1, \mathbf{e}^2, \dots, \mathbf{e}^{N \times T}\}$ (i.e., **Environment Merger** in Figure 1).

*4.1.2 Environment-aware Generation.* Upon receiving the global environment message $E = \{\mathbf{e}^1, \mathbf{e}^2, \dots, \mathbf{e}^{N \times T}\}$, for each client $C_k$, we train an *environment-aware generator* $\mathbb{EG}(\cdot)$ to transform the local rationale representation $\mathbf{h}_{r_k}^i$ to a novel environment distribution, conditioned on the novel environment message $\mathbf{e}^j$:

$$\mathbf{h}_{r_k}^j = \mathbb{EG}\left(\mathbf{h}_{r_k}, \mathbf{e}^j\right), \tag{6}$$

where $\mathbf{e}^j$ is randomly sampled from $E$. In practical implementation, we define $\mathbb{EG}(\cdot)$ as the addition function (i.e., $\mathbf{h}_{r_k}^j = \mathbf{h}_{r_k} + \mathbf{e}^j$). Through this approach, we enable the mapping of the local rationale to other environments, thereby generating counterfactual samples and disrupting the original data distribution.

*4.1.3 Predictor.* The predictor $\Phi(\cdot)$ generates prediction results by incorporating both the original graph representations and the counterfactual ones. Importantly, it should be emphasized that the environment does not directly influence task predictions. Therefore, the labels of the counterfactual samples remain unchanged. The prediction loss with counterfactual samples can be formulated as:

$$\hat{Y}_{r_k}^j = \Phi\left(\mathbf{h}_{r_k}^j\right), \quad \mathcal{L}_{r_k}^j = \mathbb{E}_{(G_k, Y_k) \sim \mathcal{D}_k}\left[\ell(\hat{Y}_{r_k}^j, Y_k)\right]. \tag{7}$$

**Algorithm 1** Training process of EaDA

**Server Executes:**
Initialize the warm-up communication round $T_w$ as 1, the communication round $T_c$, the number of environment $T$ for each client, the epoch $Ep$, the numbers of clients $N$ and the shared global/local model $f^0(\cdot)$.
**for** each communication round $t$=1 **to** $T_w + T_c$ **do**
    **for** each client id $k$=1 **to** $N$ **in parallel do**
        **if** $t \le T_w$ **then**
            ClientUpdate($k, f_k^{t-1}(\cdot)$).
        **else**
            ClientUpdate($k, f_k^{t-1}(\cdot), f^t(\cdot), E$).
        **end if**
    **end for**
    Receive all local updated model: $\left\{ f_k^t(\cdot) \right\}_{k=1}^N$, and inferred environments: $\{E_k\}_{k=1}^N$.
    Perform aggregation by Eq(1) to get $f^{t+1}(\cdot)$.
    Merge the inferred environments to achieve the global environment information $E = \{\mathbf{e}^1, \mathbf{e}^2, \dots, \mathbf{e}^{N \times T}\}$.
**end for**
**ClientUpdate**($k, \ f_k^{t-1}(\cdot), \ f^t(\cdot)$=None, $E$=None):
**for** epoch $e$=1 **to** $Ep$ **do**
    **if** $f^t(\cdot)$ is None **then**
        Update local model by Eq(4).
    **else**
        # *The environment-aware rationale extraction module.*
        1. Employ the prototype learning methods to infer the environments $E_k$ based on Eq(5).
        2. Generate the counterfactual samples $\mathbf{h}_{r_k}^j$ based on both $\mathbf{h}_{r_k}$ and environments $E$ with Eq(6).
        3. Yield task results based on both original and counterfactual samples with Eq(2) and Eq(7).
        # *The local-global alignment module.*
        4. Align the local rationale representations with the global ones based on Eq(8).
        5. Update local model with the two modules by Eq(9).
    **end if**
**end for**

## 4.2 Local-Global Alignment

In addressing the challenge posed by data heterogeneity, inherent to federated learning, we further introduce a local-global alignment module. This module aims to align global rationale representations with their local counterparts through a contrastive learning approach. By integrating global information into local training, we effectively mitigate the local shortcut problem exacerbated by data heterogeneity. Specifically, within client $C_k$, we employ the following contrastive loss:

$$\mathcal{L}_c^k = -\log \frac{\exp\left(\mathbf{h}_{r_k}^\top \mathbf{h}_r / \tau\right)}{\exp\left(\mathbf{h}_{r_k}^\top \mathbf{h}_r / \tau\right) + \sum_{\mathbf{h}_{e_k} \in \mathcal{E}} \exp\left(\mathbf{h}_{r_k}^\top \mathbf{h}_{e_k} / \tau\right)}, \quad (8)$$

where the global rationale representation $\mathbf{h}_r$ serves as the positive sample counterpart to the local $\mathbf{h}_{r_k}$. Additionally, $\mathcal{E}$ encompasses

all complement representations within the mini-batch data. The parameter $\tau$ represents a temperature parameter governing the concentration of the distribution.

Minimizing Eq(8) enables the convergence of the global rationale and the local rationale, enhancing their alignment. Moreover, it facilitates the divergence of complement representations $\mathbf{h}_{e_k}$ from rationale representations $\mathbf{h}_{r_k}$. This divergence ensures that complement representations do not encapsulate rationale information, thereby enhancing the effectiveness of environment inference by the prototype learning-based method.

## 4.3 Training and Inference.

During the training, by recalling Eq(4) and Eq(8), the overall objective of EaDA in each client $C_k$ is defined as :

$$\mathcal{L}_{EaDA}^k = \mathcal{L}_{rat}^k + \lambda_c \mathcal{L}_c^k + \sum_{j=1}^{N \times T} \mathcal{L}_{r_k}^j. \quad (9)$$

The overall training algorithm of EaDA is presented in Algorithm 1.
In the inference phase, only $\mathbf{h}_r$ that derived by the global server is employed to yield task results.

## 5 EXPERIMENTS

In this section, to demonstrate the effectiveness of EaDA, we design experiments to address the following research questions:

- **RQ1:** How effectively does EaDA perform in terms of task prediction and rationale extraction?
- **RQ2:** For the different *components* and *hyperparameters* in EaDA, respectively, what are their impacts on performance?
- **RQ3:** How scalable is EaDA as a federated learning (FL) model?
- **RQ4:** Can EaDA help other rationalization methods to improve their performance?

## 5.1 Datasets

*5.1.1 Synthetic Dataset.* In this paper, we employ the Spurious-Motif dataset [39, 44] as a synthetic benchmark for motif type prediction. Each graph in the dataset contains two subgraphs: the motif subgraph $R$ and the base subgraph $C$. Among them, $R$ serves as the rationale for motif type prediction, including three types: *Cycle*, *House*, and *Crane*, denoted as $R = \{0, 1, 2\}$. Conversely, $C$ varies according to the motif type and acts as a complement, consists of three types: *Tree*, *Ladder*, and *Wheel*, represented as $E = \{0, 1, 2\}$. Therefore, a graph in Spurious-Motif can be shown as *House-Tree*. To introduce the shortcuts into this benchmark, during dataset construction, the motif subgraph is sampled uniformly, while the base subgraph is selected based on the probability $P(C) = b \times \mathbb{I}(C = R) + \frac{1-b}{2} \times \mathbb{I}(C \neq R)$, where $b$ controls the degree of shortcut presence, with higher values indicating more pronounced shortcuts. This study considers three datasets with $b = \{0.5, 0.7, 0.9\}$. To ensure a fair evaluation, a de-biased (balanced) dataset is constructed for the test set by setting $b = \frac{1}{3}$.

*5.1.2 OGB.* In this paper, for real-world applications, we make experiments on the Open Graph Benchmark (OGB) [18], including MolHIV, MolToxCast, MolBBBP, MolBACE, and MolSIDER. To ensure a fair evaluation, we initially adopt the default scaffold splitting

**Table 1: Performance on the Synthetic Dataset and Real-world Dataset in FL scenarios.**

| | Spurious-Motif (ACC) | | | OGB (AUC) | | | | |
| --- | --- | --- | --- | --- | --- | --- | --- | --- |
| | bias=0.5 | bias=0.7 | bias=0.9 | MolHIV | MolToxCast | MolBBBP | MolSIDER | MolBACE |
| Fed-vanillaGR | 0.3182 ± 0.0353 | 0.3681 ± 0.0359 | 0.3031 ± 0.0291 | 0.6985 ± 0.0155 | 0.6111 ± 0.0055 | 0.6339 ± 0.0142 | 0.5774 ± 0.0175 | 0.7058 ± 0.0334 |
| FedDisC | 0.4418 ± 0.0182 | 0.4481 ± 0.0381 | 0.3579 ± 0.0471 | 0.7212 ± 0.0201 | 0.6274 ± 0.0018 | 0.6561 ± 0.0121 | 0.5869 ± 0.0142 | 0.7253 ± 0.0290 |
| FedCAL | 0.4213 ± 0.0109 | 0.5289 ± 0.0087 | 0.4191 ± 0.0248 | 0.7039 ± 0.0113 | 0.6170 ± 0.0051 | 0.6575 ± 0.0076 | 0.5879 ± 0.0138 | 0.7248 ± 0.0212 |
| FedGSAT | 0.4281 ± 0.0328 | 0.5259 ± 0.0381 | 0.4194 ± 0.0338 | 0.7149 ± 0.0226 | 0.6255 ± 0.0030 | 0.6555 ± 0.0085 | 0.5952 ± 0.0082 | 0.7369 ± 0.0413 |
| FedDARE | 0.4483 ± 0.0193 | 0.4891 ± 0.0391 | 0.4288 ± 0.0977 | 0.7220 ± 0.0165 | 0.6289 ± 0.0059 | 0.6621 ± 0.0096 | 0.5886 ± 0.0113 | 0.7301 ± 0.0092 |
| FedRGDA | 0.4087 ± 0.0293 | 0.5089 ± 0.0198 | 0.4286 ± 0.0313 | 0.7246 ± 0.0085 | 0.6235 ± 0.0034 | 0.6605 ± 0.0157 | 0.5906 ± 0.0151 | 0.7282 ± 0.0301 |
| FedGR | 0.4610 ± 0.0289 | 0.5538 ± 0.0398 | 0.4977 ± 0.0315 | 0.7387 ± 0.0186 | 0.6316 ± 0.0054 | 0.6690 ± 0.0174 | 0.6017 ± 0.0202 | 0.7435 ± 0.0170 |
| EaDA | **0.5269 ± 0.0273** | **0.5892 ± 0.0163** | **0.5447 ± 0.0365** | **0.7611 ± 0.0084** | **0.6345 ± 0.0108** | **0.6713 ± 0.0077** | **0.6178 ± 0.0040** | **0.7743 ± 0.0073** |
| EaDA-ERE | 0.4344 ± 0.0138 | 0.5276 ± 0.0121 | 0.4302 ± 0.0288 | 0.7123 ± 0.0034 | 0.6148 ± 0.0025 | 0.6522 ± 0.0047 | 0.5882 ± 0.0032 | 0.7334 ± 0.0056 |
| EaDA-LDA | 0.4824 ± 0.0348 | 0.5677 ± 0.0225 | 0.5011 ± 0.0426 | 0.7536 ± 0.0164 | 0.6301 ± 0.0202 | 0.6667 ± 0.0122 | 0.6032 ± 0.0092 | 0.7597 ± 0.0230 |

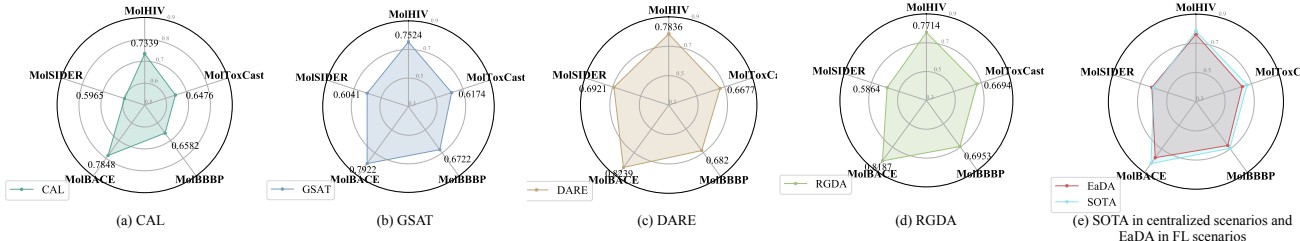

(a) CAL            (b) GSAT            (c) DARE            (d) RGDA            (e) SOTA in centralized scenarios and EaDA in FL scenarios

**Figure 2: (a)-(d): Performance of rationalization methods in _centralized scenarios_. (e): The upper bound of FL methods, where the SOTA results in centralized scenarios can be considered as the upper bound of rationalization in _FL scenarios_.**

method in OGB to partition the datasets into training, validation, and test sets. Notably, under this scaffold-based partition, the distribution of the test and training sets significantly differs, indicating different environments. In essence, the test set is out-of-distribution relative to the training set.

Considering that the above datasets are all standard centralized datasets, we employ the following method to partition the training set across various clients to conform to the settings of FL scenarios. Specifically, we distribute the constructed training dataset to $N$ clients using the unbalanced partition algorithm Latent Dirichlet Allocation (LDA) [16, 17]. This approach involves generating a heterogeneous partition by sampling $p_i \sim \text{Dir}_N(\gamma)$, thereby allocating a proportion $p_{i,n}$ of training instances for class $i$ to each local client. In this paper, for Spurious-Motif, $N$ is set to 3 and $\gamma$ to 3. For OGB, we set $N$ to 4 and $\gamma$ to 4.

We also explore alternative dataset partitioning methods and present the corresponding experimental results in section 5.6. The comprehensive dataset description is available in Appendix A.

## 5.2 Comparison Methods

In this section, to validate the effectiveness of EaDA, we first compare our method with several rationale-based methods: DisC [9], GSAT [28], CAL [34], DARE [45], and RGDA [23]. These methods are adapted from centralized scenarios to FL scenarios by implementing them within the Fed-vanillaGR framework. We name these adaptations as FedDisC, FedGSAT, FedCAL, FedDARE, and FedRGDA, respectively. Besides, we also compare EaDA with FedGR [46], which is the first federated graph rationalization method. Furthermore, we implement two variants of EaDA: EaDA without the environment-aware rationale extraction module (EaDA-ERE)

**Table 2: Training speed of federated graph rationalizations.**

| Methods | Training Speed |
| --- | --- |
| Fed-vanillaGR | 1.00 × |
| FedGR | 4.72 × |
| EaDA | 1.11 × |

and EaDA without the local-global alignment module (EaDA-LGA). Details of the comparison methods are shown in Appendix B.

## 5.3 Experimental Setup

During the evaluation phase, the AUC metric is utilized to assess the task prediction performance of OGB and ACC is used in Spurious-Motif. Since Spurious-Motif includes ground-truth rationales, we can evaluate the precision of the extracted rationales on Spurious-Motif with the Precision@5 metric. This metric measures the accuracy of the top 5 extracted rationales by comparing them to the ground-truth rationales. All methods, including the EaDA approach and other comparison methods, undergo training on a single A100 GPU with 5 different random seeds. The reported test performance comprises the mean results and standard deviations acquired from the epoch that attains the highest validation prediction performance. Several results of comparison methods in Table 1 are directly taken from [46]. Detailed experimental and hyperparameter setups can be found in Appendix C.

## 5.4 Overall Performance (RQ1)

*5.4.1 Performance of the Task Prediction.* To evaluate the effectiveness of EaDA, a comparative analysis is conducted against various

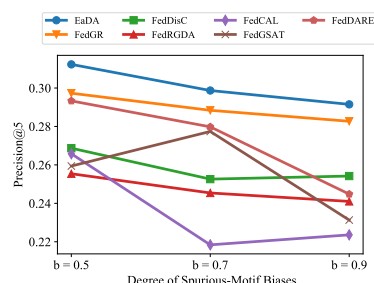

**Figure 3: Results of Precision@5 between extracted rationales and the ground-truth rationales on Spurious-Motif.**

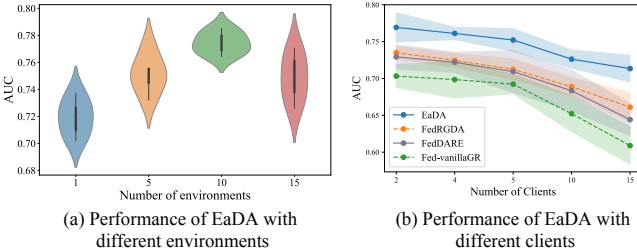

(a) Performance of EaDA with different environments

(b) Performance of EaDA with different clients

**Figure 4: (a) Varying the specified environment number $k$ on MolBACE. (b) Performance of EaDA with different number of clients on MolBACE.**

baseline methods in the task prediction and the experimental results are shown in Table 1 and Figure 2. Specifically, in Table 1, For the rationalization baselines transferred from the centralized scenario (e.g. FedCAL and FedRGDA), their performances are not bad, and all of them are significantly improved compared to Fed-vanillaGR. However, they don't perform as good as FedGR and EaDA, where both FedGR and EaDA are rationalization methods specifically designed for FL scenarios. Compared to the state-of-the-art (SOTA) method, FedGR, our model performs better across all datasets, demonstrating the effectiveness of our prototype learning-based environment inference method. Meanwhile, in Table 2, we present the results of our experiments comparing the training speed of EaDA with FedGR. The hardware setup for the experiments consists of 12 cores of Intel(R) Xeon(R) Gold 5317 CPU and a single 40G NVIDIA A100 Tensor Core GPU. From the results, we find EaDA achieves an impressive training speed, approximately **4.25 times** the speed achieved by FedGR. This further illustrates the necessity of EaDA's assumption that the environment can be inferred, compared to FedGR's assumption of an unavailable environment.

*5.4.2 Performance of the Rationale Extraction.* Furthermore, to examine whether EaDA avoids composing rationales by extracting shortcuts, we conduct experiments on Spurious-Motif, which includes ground-truth rationales. In Figure 3, we can observe that as the degree of bias in the Spurious-Motif dataset changes, our method consistently outperforms the baseline methods. This indicates that EaDA effectively addresses data heterogeneity issues and the exacerbated problem of local shortcut learning, enhancing the faithfulness of rationale extraction.

*5.4.3 Experimental Upper Bound of EaDA.* In Figure 2, we also report the performance of the rationalization baseline methods

in the ***centralized scenario*** across datasets. Meanwhile, we also extract the optimal performance on each dataset and show it as the SOTA results under the centralized scenario in Figure 2(e). This result can be considered as the upper bound of the capability that the method can achieve in the FL scenario. By comparing with EaDA (The red line in Figure 2 (e)), we find that EaDA is very close to this upper bound of capability on several datasets (our approach even exceeds the bound in MolSIDER), illustrating the necessity of introducing global information into local training, and the effectiveness of the ERE and LDA modules that we designed.

## 5.5 Ablation Study (RQ2)

In this subsection, we validate the efficacy of each module proposed in our paper and provide primary ablation studies for all of them. Firstly, we compare EaDA with EaDA-LDA. In EaDA-LDA, we remove the LDA module, making it impossible to align the global rationale with the local rationale information. As shown in Table 1, this results in a decrease in the effectiveness of EaDA-LDA. Additionally, due to the absence of the contrastive learning loss constraint, the separated rationale and complement representations are not fully disentangled. Consequently, it becomes possible for the complement information to contain some of the rationale information. This leads to inaccurate environment inference using the prototype learning-based method, affecting the aggregation of global information and consequently impacting overall performance. This analysis underscores the necessity of designing the LDA module. Subsequently, we compare EaDA with EaDA-ERE. From Table 1, it is evident that the effectiveness of EaDA-ERE decreases more significantly, indicating that the ERE module is more crucial compared to the LDA module. Without information about the environment, relying solely on the alignment of global rationale and local rationale fails to break the spurious correlation between the environment and labels in the data. Consequently, mitigating the shortcut problem becomes challenging.

## 5.6 Sensitivity Analysis (RQ2)

*5.6.1 EaDA with Different Dataset Partitioning Methods.* In this subsection, we initially investigate the sensitivity of EaDA to different federated dataset partitioning methods. To achieve this, we deliberately create more unbalanced data distributions in our experiments. For MolHIV, MolBBBP, MolBACE, all being binary classification datasets where the labels are either 0 or 1, we intentionally create class-unbalanced distributions among clients. The specific data partitioning is depicted in Table 7. From the table, it's evident that after the repartitioning, the ratios of positive and negative categories among the different clients exhibit significant variation. Additionally, for MolToxCast and MolSIDER, both being multi-label classification datasets with numerous categories, we opt to partition the dataset based on the number of nodes in each graph. Specifically, graphs with fewer nodes are assigned to one client, while those with more nodes are allocated to another client. The details of this data partitioning are presented in Table 7. Observing the table, we note a substantial discrepancy in the number of nodes/edges among graphs across different clients, indicating an unbalanced distribution among clients. Finally, utilizing the repartitioned datasets, we conduct experiments with EaDA and other baseline methods.

**Table 3: Performance on the Real-world Dataset in FL scenarios with another partition method.**

|  | MolHIV | MolToxCast | MolBBBP | MolSIDER | MolBACE |
|---|---|---|---|---|---|
| Fed-vanillaGR | 0.6877 ± 0.0180 | 0.5943 ± 0.0042 | 0.6232 ± 0.0118 | 0.5313 ± 0.0118 | 0.6832 ± 0.0248 |
| FedDisC | 0.7102 ± 0.0031 | 0.6082 ± 0.0031 | 0.6438 ± 0.0048 | 0.5423 ± 0.0193 | 0.7088 ± 0.0234 |
| FedCAL | 0.6987 ± 0.0130 | 0.5985 ± 0.0058 | 0.6489 ± 0.0032 | 0.5489 ± 0.0024 | 0.7123 ± 0.0387 |
| FedGSAT | 0.7083 ± 0.0034 | 0.6055 ± 0.0046 | 0.6518 ± 0.0024 | 0.5573 ± 0.0137 | 0.7177 ± 0.0303 |
| FedDARE | 0.6829 ± 0.0177 | 0.6021 ± 0.0049 | 0.6482 ± 0.0083 | 0.5498 ± 0.0294 | 0.7003 ± 0.0205 |
| FedRGDA | 0.7031 ± 0.0035 | 0.5953 ± 0.0060 | 0.6502 ± 0.0095 | 0.5512 ± 0.0078 | 0.7276 ± 0.0320 |
| FedGR | 0.7290 ± 0.0061 | 0.6179 ± 0.0159 | 0.6654 ± 0.0121 | 0.5697 ± 0.0028 | 0.7743 ± 0.0145 |
| **EaDA** | **0.7308 ± 0.0243** | **0.6206 ± 0.0104** | **0.6748 ± 0.0184** | **0.5735 ± 0.0039** | **0.7636 ± 0.0093** |

**Table 4: Structural Generalizability of the ERE module. Each rationalization method with ERE is highlighted in gray.**

|  | MolHIV | MolToxCast | MolBBBP | MolSIDER | MolBACE |
|---|---|---|---|---|---|
| FedGSAT | 0.7149 | 0.6255 | 0.6555 | 0.5952 | 0.7369 |
| +ERE | 0.7490 (↑3.41%) | 0.6023 (↓2.32%) | 0.6693 (↑1.38%) | 0.5994 (↑0.42%) | 0.7543 (↑1.74%) |
| FedDARE | 0.7220 | 0.6289 | 0.6621 | 0.5886 | 0.7301 |
| +ERE | 0.7511 (↑2.91%) | 0.6304 (↑0.15%) | 0.6704 (↑0.83%) | 0.5904 (↑0.18%) | 0.7539 (↑2.38%) |
| FedRGDA | 0.7246 | 0.6235 | 0.6605 | 0.5906 | 0.7282 |
| +ERE | 0.7658 (↑4.12%) | 0.6287 (↑0.52%) | 0.6693 (↑0.88%) | 0.5803 (↓1.03%) | 0.7431 (↑1.49%) |

The experimental results are summarized in Table 3. From these experimental results, it is evident that EaDA consistently achieves optimal performance even when the client dataset partitioning method is altered. This experiment underscores the versatility of our approach, demonstrating its efficacy across various federated learning scenarios with differing data distributions.

*5.6.2 EaDA with the Different Number of Inferred Environments.* We conduct parameter sensitivity experiments on the number $k$ of inferred environments in EaDA. The number of environments is crucial for subsequent counterfactual generation methods and forms the basis for mitigating shortcuts in our ERE method. Properly choosing $k$ is essential for the effectiveness of the model, as it determines the granularity of the environment partitioning and impacts the ability to generate diverse counterfactual samples. To explore the sensitivity of our methods to the parameter $k$, we vary $k$ and observe its impact on performance. Figure 4(a) illustrates the performance of EaDA with different environment numbers $k$. The figure clearly shows that the performance of our methods deteriorates when $k$ is too small (e.g., $k = 1$) or too large (e.g., $k = 15$). When $k = 1$, all training data are considered to be from a single environment. This results in the poorest performance. The inability to partition the training samples into multiple environments means that the model fails to capture the underlying variations and spurious correlations present in the data. Consequently, the counterfactual generation is less effective, leading to suboptimal performance. Besides, when $k$ is too large (e.g., $k = 15$), the performance also deteriorates. This can be attributed to over-partitioning the training data into too many environments, leading to fragmentation and difficulty in learning meaningful environment.

## 5.7 Scalability Analysis (RQ3)

Scalability is a crucial consideration in federated learning, and understanding how a method performs under increasing numbers of clients is essential. Therefore, we explore how EaDA scales with an increasing number of clients. As depicted in Figure 4(b), the effectiveness of EaDA and other rationalization baseline methods diminishes as the number of clients increases. However, even as the number of clients grows, EaDA consistently outperforms the other methods, maintaining a high AUC metric. For instance, at $N = 15$ clients, EaDA exhibits only a 4.77% decrease in effectiveness compared to its performance at $N = 4$, while Fed-vanillaGR experiences an 8.96% decrease. These results highlight EaDA's superior scalability in accommodating an increasing number of clients. The ability of EaDA to maintain its performance with an increasing number of clients can be attributed to its robust design. The method incorporates both the environment-aware rationale extraction (ERE) module and the local-global alignment (LDA) module, which together ensure effective handling of data heterogeneity and alignment of global and local information. This design mitigates the adverse effects of data fragmentation and distributional discrepancies that typically arise in federated learning scenarios with numerous clients.

## 5.8 Structural Generalizability of EaDA (RQ4)

Through ablation experiments, we find that our Environment-aware Rationale Extraction (ERE) module significantly enhances EaDA's performance. This observation raises an intriguing research question: *Can our ERE module enhance the performance of other rationale-based methods in federated learning scenarios?* To address this question, we integrate the ERE module into FedGSAT, FedCAL, Fed-DARE, and FedRGDA and analyze the corresponding results. From Table 4, it's evident that incorporating our ERE module consistently improves the performance of all rationale-based methods. This finding suggests that our ERE module exhibits generalizability and scalability, effectively enhancing the performance of other rationale-based methods in federated learning scenarios.

## 6 CONCLUSION

In this paper, we proposed an Environment-aware Data Augmentation (EaDA) method for Federated Graph Rationalization, addressing challenges in data heterogeneity and the local shortcut problem. This method comprised two key components: the Environment-aware Rationale Extraction (ERE) module and the Local-Global Alignment (LGA) module. The ERE module inferred and shared abstracted environmental information among clients, allowing for the generation of counterfactual samples to compose faithful rationales. The LGA module employed the contrastive learning method to align local and global rationales, mitigating data heterogeneity. Our method exhibited enhanced effectiveness compared to existing rationalization approaches, as demonstrated through experiments on both real-world and synthetic datasets.

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

## A  DATA STATISTICS

To demonstrate the effectiveness of EaDA, we conduct experiments several datasets. Specifically, for synthetic dataset, we use the Spurious-Motif [39, 44]. For real-world dataset, we utilize the Open Graph Benchmark (OGB) [18] and focus on the OGB-Mol datasets available within OGB, including MolHIV, MolToxCast, MolBBBP, MolBACE and MolSIDER, which provide diverse molecular properties for analysis and prediction. Details of dataset statistics are summarized in Table 5 and Table 6.

Besides, Table 7 shows the dataset statistics for each client after dividing the OGB using other dataset partitioning methods.

## B  COMPARISON METHODS

In this section, we present the details of our comparison methods:

- **DisC** [9]: Employs a disentangling method to learn causal and shortcut substructures within graph data. By synthesizing counterfactual training samples, DisC aims to further de-correlate causal and shortcut variables, thereby mitigating the influence of shortcuts.
- **GSAT** [28]: Introduces stochasticity to block label-irrelevant information and selectively identifies label-relevant subgraphs, guided by the information bottleneck principle [1, 38].
- **CAL** [34]: Proposes the Causal Attention Learning (CAL) strategy, which composes causal rationales and mitigates the confounding effect of shortcuts to achieve high generalization.
- **DARE** [45]: Introduces a self-guided method with a disentanglement operation to encapsulate more information from the input and extract rationales.
- **RGDA** [23]: Generates counterfactual samples using the bias substructure, but lacks a disentanglement operation to ensure the bias can be separated from the original input.
- **FedGR** [46]: Designs a difference-aware data augmentation method to generate shortcut-conflicted samples for each client by assuming the client environment is unavailable.
- **EaDA-ERE**: A variant of EaDA that removes the environment-aware rationale extraction module. The objective of EaDA-ERE is degraded from Eq(9) to $\mathcal{L}_{ere^-}^k = \mathcal{L}_{rat}^k + \lambda_c \mathcal{L}_c^k$.
- **EaDA-LDA**: Achieved by excluding the local-global alignment module from EaDA. Its objective is $\mathcal{L}_{lda^-}^k = \mathcal{L}_{rat}^k + \sum_{j=1}^{n \times k} \mathcal{L}_{r_k}^j$.

## C  EXPERIMENTAL SETUPS

In all experimental settings, both the values of the hyperparameters $\lambda_{sp}$ and $\lambda_c$ are set to 1.0. The hidden dimensionality $d$ is 128 for the OGB dataset and 32 for Spurious-Motif dataset. During the training process, we employ the Adam optimizer [20] with a learning rate initialized as 1e-3 for the OGB and 1e-2 for Spurious-Motif. The number of the inferred environment for each client is set to 5 for MolHIV and MolToxCast, and 10 for other OGB datasets, 3 for Spurious-Motif. Following [46], we set the predefined sparsity $\alpha$ as 0.1 for MolHIV, 0.5 for MolSIDER, MolToxCast, MolBBBP and MolBACE, and 0.4 for other datasets. The communication round $T_c$ is 20 and the epoch in each communication is 10, totaling 200 iterations. In this study, we employ GIN as the backbone to implement both our methods and comparison methods.

## D  CASE STUDY

In this section, we make experiments on Spurious-Motif ($b = 0.9$) to show the rationales extracted by EaDA. In Figure 5, each graph comprises a motif type (such as *Cycle*, *House*, and *Crane*) and a base (like *Tree*, *Wheel*, and *Ladder*). The highlighted navy blue nodes indicate selected rationale nodes[2]. If there is an edge between two identified nodes, it is visualized as a red line. From the figure, we can find that EaDA effectively extracts more accurate rationales for prediction, enhancing the model's explainability and overall performance.

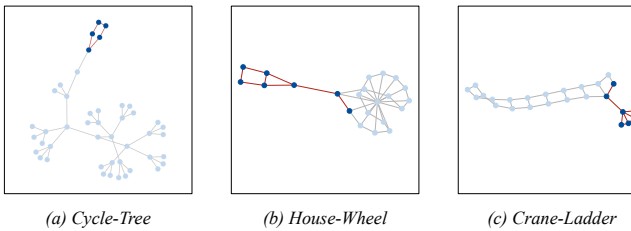



(a) Cycle-Tree     (b) House-Wheel     (c) Crane-Ladder

**Figure 5: Visualization of EaDA rationale subgraphs.**



---

[2]In this paper, when the probability of predicting a node as part of rationales $\tilde{m}_i$ exceeds 0.55, we consider the node as part of the rationales.

**Table 5: Statistics of Spurious-Motif Datasets. Among them, different clients share the same valid and test set.**

| | Spurious-Motif | | |
| --- | --- | --- | --- |
| | b=0.5 | b=0.7 | b=0.9 |
| Client1/Client2/Client3/Val/Test | 377/662/1961/3,000/6,000 | 377/662/1,961/3,000/6,000 | 377/662/1,961/3,000/6,000 |
| Classes | 3 | 3 | 3 |
| Avg. Nodes | 18.60/18.29/18.48/18.50/88.80 | 18.73/18.27/18.8/18.50/88.80 | 19.02/18.54/18.66/18.50/88.80 |
| Avg. Edges | 27.72/27.31/27.55/27.54/125.14 | 28.29/27.3/28.05/27.54/125.14 | 28.74/27.63/27.81/27.54/125.14 |

**Table 6: Statistics of OGB Datasets.**

| | MolHIV | MolToxCast | MolBACE |
| --- | --- | --- | --- |
| Client1/Client2/Client3/Client4/Val/Test | 9,380/6,148/10,113/7,260/4,113/4,113 | 871/614/3,819/1,556/858/858 | 425/234/191/360/151/152 |
| Classes | 2 | 617 | 2 |
| Avg. Nodes | 25.31/25.32/25.15/25.27/27.79/25.27 | 16.41/16.86/16.63/16.91/26.17/28.19 | 33.81/33.91/33.28/33.33/37.23/34.82 |
| Avg. Edges | 54.19/54.2/53.89/54.15/61.05/55.59 | 32.91/33.93/33.45/33.99/56.09/60.71 | 73.06/73.13/71.87/72.09/81.3/75.11 |
| | MolBBBP | MolSIDER | |
| Client1/Client2/Client3/Client4/Val/Test | 472/299/325/535/204/204 | 422/333/201/185/143/143 | |
| Classes | 2 | 27 | |
| Avg. Nodes | 22.44/22.15/22.34/22.81/33.20/27.51 | 28.85/30.96/30.97/29.7/43.24/53.27 | |
| Avg. Edges | 48.42/47.53/48.05/49.19/71.84/59.75 | 60.53/64.77/64.87/62.25/91.85/112.66 | |

**Table 7: Statistics of OGB Datasets with an another partitioning method.**

| | MolHIV | MolBBBP | MolBACE |
| --- | --- | --- | --- |
| Client1/Client2/Client3/Val/Test | 1,000/30,500/1,401/4,113/4,113 | 1,062/200/369/204/204 | 900/150/160/151/152 |
| Class Ratio (Positive(1): Negative(0)) among clients | 500:500/500:30,000/232:1,169 | 1,000:62/100:100/269:100 | 300:600/100:50/80:80 |
| Avg. Nodes | 25.02/25.03/30.34/27.79/25.27 | 21.56/23.36/24.69/33.20/27.51 | 34.18/34.12/29.86/37.23/34.82 |
| Avg. Edges | 52.80/53.61/65.60/61.05/55.59 | 46.43/50.03/53.33/71.84/59.75 | 73.83/73.61/64.74/81.3/75.11 |
| | MolToxCast | MolSIDER | |
| Client1/Client2/Client3/Val/Test | 3,000/2,000/1,860/858/858 | 400/500/241/143/143 | |
| Classes | 617 | 27 | |
| Avg. Nodes | 9.42/16.32/28.79/26.17/28.19 | 12.66/25.53/67.95/43.24/53.27 | |
| Avg. Edges | 17.73/32.92/59.75/56.09/60.71 | 24.96/54.86/142.12/91.85/112.66 | |

