# OpenReview forum: "Empowering Federated Graph Rationale Learning with Latent Environments"
_ACM.org/TheWebConf/2025/Conference — WWW 2025 Poster_

### Official Review · Reviewer_vsAj · 2024-11-22

**Novelty:** 4
**Technical Quality:** 3

**Review:**

This study proposes the Environment-aware Data Augmentation (EaDA) method, which assumes that latent environments can be inferred. This is a novel perspective in existing research on Federated Learning (FL) and Graph Neural Networks (GNNs). The EaDA method includes the Environment-aware Rationale Extraction (ERE) module and the Local-Global Alignment (LGA) module, which clearly separate different functionalities, making the approach easier to understand and implement. By inferring environment information using prototype learning and uploading it, the method ensures client data privacy, which is particularly significant in the current context of increasing emphasis on data privacy. The LGA module mitigates performance degradation caused by data heterogeneity through contrastive learning methods to align local and global rationale representations.



**Weaknesses:**

1. The EaDA method comprises multiple modules (ERE and LGA), which could lead to increased model complexity. In practical applications, complex models may face challenges such as prolonged training times and high computational resource consumption, particularly when client resources are limited. While EaDA demonstrates certain advantages in addressing graph rationalization problems in Federated Learning, there is room for improvement in handling data heterogeneity, experimental comprehensiveness, and model complexity.

2. The paper highlights that EaDA’s performance is affected by the choice of the number of environments (k). When k is too small or too large, the model performance declines. This suggests challenges in selecting the appropriate number of environments, which may prevent the model from effectively capturing variations and correlations in the data. This sensitivity could limit the model's adaptability to different data distributions.

**Questions:**

1. When addressing the problem of graph rationalization, do the ERE and LDA modules in EaDA effectively avoid biases caused by data heterogeneity? Specifically, does EaDA demonstrate clear experimental results showing superiority over other methods in generating aligned representations through contrastive learning and eliminating local shortcut problems? In comparison with other rationale-based methods (e.g., FedDisC, FedGSAT), what specific advantages does EaDA exhibit, particularly in rationale extraction and model interpretability?

2. In the ablation study, the two modules of EaDA (LDA and ERE) were compared, and the results showed that the ERE module has a greater impact on the effectiveness of the model. This may indicate that the model's design relies too heavily on certain modules, and if the performance of one module is suboptimal, it could significantly affect the overall performance. Does the team have plans to further optimize the LDA module to reduce the strong reliance on the ERE module? Could additional auxiliary mechanisms or modules be introduced to make the model more flexible in adapting to different scenarios?

3. Although EaDA performs well in handling data heterogeneity, the distribution and characteristics of data in federated learning scenarios could be more complex. The paper does not discuss in detail how the model could be further optimized to address more complex data heterogeneity problems, which could be a potential area for improvement. For more extreme distribution differences (e.g., completely unrelated datasets between clients), would the ERE and LDA modules still be applicable? Would it be necessary to introduce more dynamic adjustment mechanisms (e.g., automatically selecting the optimal number of environments (k)) during training to enhance the model's adaptability?

4. Besides metrics like AUC and Precision@5, could additional evaluation metrics (such as runtime efficiency and model convergence rate) be included to further demonstrate the advantages of the method?

**Reviewer Confidence:**

3: The reviewer is confident but not certain that the evaluation is correct

**Scope:**

4: The work is relevant to the Web and to the track, and is of broad interest to the community

---

### Official Review · Reviewer_8TX1 · 2024-11-29

**Novelty:** 3
**Technical Quality:** 3

**Review:**

Summary:

This paper investigates the federated graph rationalization problem that explores the rationalization part of graphs for prediction in the federated learning manner. To address this problem, the authors propose an environment-aware data augmentation (EaDA) method which consists of the environment-aware rationale extraction (ERE) module and the local-global alignment (LGA) module.


Strength:

1.	This paper provides a clear background and preliminary on the investigated problem.

2.	The illustration is well-designed to demonstrate the design of the proposed framework EaDA, which follows a concise logic.

3.	The proposed method is extensively evaluated in the experiments in terms of various aspects.



Weakness:

1.	The overall contribution of this paper seems restrictive. The core components of the proposed frameworks are environment rationale extraction and local-global alignment. For the former design, I wonder about the fundamental innovation compared with the existing design introduced in the Preliminary part.

2.	In the design of the environment merge, please justify the motivation for combining the non-rational part for environment information.

3.	As claimed in the Introduction part, FedGR incurs significant efficiency bottlenecks, i.e., the training time is around 5 times longer than the vanilla federated graph rationalization. However, the training efficiency of the proposed method in Section 4.3 is missing. I would suggest the authors clarify the design for efficiency enhancement.

**Questions:**

Please refer to the weakness part.

**Reviewer Confidence:**

2: The reviewer is willing to defend the evaluation, but it is likely that the reviewer did not understand parts of the paper

**Scope:**

3: The work is somewhat relevant to the Web and to the track, and is of narrow interest to a sub-community

---

### Official Review · Reviewer_1FTL · 2024-12-03

**Novelty:** 6
**Technical Quality:** 7

**Review:**

This paper introduces a novel environment-aware data augmentation method, which addresses the data heterogeneity and the impact of local shortcuts on Federation Graph Rationalization.

Pros: The whole presentation is really nice, with comprehensive descriptions and detailed analysis of the experimental design.

Cons: The authors claimed that the environmental context refers to the data distributions of specific clients; the global environment learned from the clients contains the details of data distributions across clients, which faces the risks of privacy leakage while there have misbehaved clients.

**Questions:**

In my view, federated graph rationalization contributes to the explainable AI with federated learning using meaningful (informative) subgraphs to explain the prediction results. I suggest that the authors modify the description in the introduction, providing proper significance of federated graph rationalization and summarizing the major contributions that could further enhance the quality of this paper.

**Reviewer Confidence:**

4: The reviewer is certain that the evaluation is correct and very familiar with the relevant literature

**Scope:**

4: The work is relevant to the Web and to the track, and is of broad interest to the community

---

### Official Review · Reviewer_T3dP · 2024-12-03

**Novelty:** 4
**Technical Quality:** 5

**Review:**

This paper investigates the problem of generating rationales for Graph Neural Networks in a federated setting. It addresses the heterogeneity challenge across clients by proposing Environment-aware Rationale Extraction (ERE) and Local Global Alignment (LGA). ERE  allows counterfactual samples to be shared between clients as shared environment (global information). LGA applies contrastive learning to align the local and global rationales. The proposed approach is compared against multiple baselines on synthetic and real datasets and is shown to outperform the state of the art (FedGR) in terms o prediction accuracy and scalability.

**Strengths of the paper:**
+The proposed solution is clear
+The experimental results seem promising
+The paper addresses a relevant problem

**Weaknesses of the paper:**
-The contributions of the paper are unclear
-The data partitioning schemes applied seem to benefit the proposed approach
- Writing needs improvements

**Detailed comments:**

This is an interesting paper overall. I have a few concerns that I would like to address before making my final recommendation:

-Contributions: the data heterogeneity problem in federated learning is not specific to graph rationalization. The paper does not discuss how this problem is addressed for other tasks and what is different about graph rationalization that requires a novel approach.
-Data partitioning: the advantages of the proposed approach compared with the baselines seem to depend on how the data is partitioned. Some of the gains in Table 1 are already small, which means that in real settings the differences might not be significant.
-Writing: the paper has a few writing issues that need to be addressed. First, LDA and LGA are used to refer to the same thing. Significant part of Section 3 is dedicated to discussing existing solutions instead of the proposed approach. Figure 1 is quite important and deserves a more detailed caption describing the proposed method. Each table and plot in Section 5 show say what the evaluation metric considered is. Figure 2.e is a it misleading and should probably be presented as a table.

**Questions:**

1) How is heterogeneity in rationalization addressed differently than in other federated settings?

2) Why is the data partitioning approach applied in the paper reasonable?

**Reviewer Confidence:**

3: The reviewer is confident but not certain that the evaluation is correct

**Scope:**

3: The work is somewhat relevant to the Web and to the track, and is of narrow interest to a sub-community